# The Correlation between Proliferative Immunohistochemical Markers and Papillary Thyroid Carcinoma Aggressiveness

**DOI:** 10.3390/medicina59010110

**Published:** 2023-01-04

**Authors:** Mohammad Sheikh-Ahmad, Yara Shalata, Jacob Bejar, Hila Kreizman Shefer, Majd F. Sirhan, Monica Laniado, Ibrahim Matter, Abed Agbarya, Maria Reut, Ekaterina Yovanovich, Leonard Saiegh

**Affiliations:** 1Institute of Endocrinology, Bnai Zion Medical Center, 47 Golomb St., Haifa 31048, Israel; 2Department of Pathology, Bnai Zion Medical Center, 47 Golomb St., Haifa 31048, Israel; 3Department of Surgery, Bnai Zion Medical Center, 47 Golomb St., Haifa 31048, Israel; 4Department of Oncology, Bnai Zion Medical Center, 47 Golomb St., Haifa 31048, Israel

**Keywords:** papillary thyroid carcinoma, immunohistochemical markers, extrathyroidal invasion, metastases

## Abstract

*Background and Objectives:* Papillary thyroid carcinoma (PTC) is one of the most common malignancies of the endocrine system. In order to improve the ability to predict tumor behavior, several studies have been conducted to search for surrogate prognostic immunohistochemical tumor markers. Objective: To evaluate the correlation between the intensity of different immunohistochemical marker staining in PTC and the risk for extrathyroidal extension and metastases. *Materials and Methods:* The study comprised patients who underwent hemi- or total thyroidectomy. Thyroid tissues were immunohistochemically stained for different tumor proliferative markers: Minichromosome maintenance proteins 2 (MCM2), Ki-67 labeling index, E-Cadherin, Neuropilin-1 and Metallothionein. The correlation between the intensity of each marker staining and the final diagnosis (benign neoplasm and PTC) and the correlation between the intensity of each staining and tumor extrathyroidal extension and metastases were evaluated. *Results:* The study included 66 patients. Staining for Metallothionein, E-Cadherin and MCM2 significantly differed between benign neoplasm (*n* = 22) and thyroid-confined PTC (*n* = 21) (*p* = 0.002, 0.004 and 0.005, respectively), between benign neoplasm and PTC with extrathyroidal extension (*n* = 11) (*p* = 0.001, 0.006 and 0.01, respectively), and between benign neoplasm and PTC with metastases (*n* = 12) (*p* = 0.01, <0.001 and 0.037, respectively). No staining correlated with extrathyroidal extension. The intensity of E-Cadherin staining was significantly lower in PTC with metastases than thyroid confined PTC and PTC with extrathyroidal extension (*p* = 0.028 and 0.021, respectively). *Conclusions:* Immunohistochemical staining for Metallothionein, E-Cadherin and MCM2 significantly distinguished between benign thyroid tumor and PTC. E-Cadherin staining significantly and inversely correlated with the presence of metastases.

## 1. Introduction

Thyroid carcinoma is the most common cancer of the endocrine system, and its prevalence has increased in recent years [1]. Papillary thyroid carcinoma (PTC) accounts for 85% to 90% of thyroid malignancies and is characterized by slow growth and an excellent prognosis [1]. Still, in some cases, PTC can be invasive and may be associated with cervical lymph node metastases (CLNM), distant metastases (DM), and death [2]. In order to identify high-risk PTC cases, several studies have been conducted to assess tumor aggressiveness using different histopathological proliferative markers.

A few studies have explored the correlation between immunohistochemical staining of PTC and the risk of tumor metastases and extrathyroidal extension (ETE). Ki-67 labeling index (Ki-LI) is a tumor proliferation marker that is widely used for immunohistochemical staining and has been shown to be useful as a prognostic marker in several cancers [3]. Still, immunohistochemistry using Ki-LI in PTC did not show consistent results. Lee et al. found no correlation between Ki-LI, tumor size or ETE in 60 cases of PTC [4], while others did report a positive correlation between Ki-LI and risk of ETE and metastases [5,6].

Metallothionein (MT), a cysteine-rich protein, is a transcription factor overexpressed in various tumors. Wang et al. [7] showed in their study on 129 cases of PTC, that MT overexpression is correlated with high TNM tumor stage (stages III-IV) and the presence of CLNM.

Another tumor proliferation marker is E-Cadherin, an epithelial trans-membrane protein that plays a role in cell adhesion, which when losing its expression, may be associated with an increased risk of tumor metastases [8]. In one study, 100% of PTC tumors with positive expression of E-cadherin had no metastases, and positive staining was demonstrated in 80% of TNM stage I and II tumors, and in 50% of stage III and IV tumors [8]. These data indicate an inverse correlation between positive staining of E-Cadherin and risk of PTC metastases.

Neuropilin-1 (NP-1), another immunohistochemical marker used for the grading of some malignant tumors, has not yet been studied in PTC. NP-1 is a transmembrane glycoprotein of epithelial cells, and it serves as a co-receptor for vascular endothelial growth factors (VEGFs) and tyrosine kinase. NP-1 takes part in several cellular processes, as angiogenesis, cell migration, vascularization and cancer progression [9]. NP-1 overexpression is evident in brain, prostate, breast, colon and lung tumors [9]. Moreover, strong positive histochemical staining of NP-1 in some malignancies, such as nasopharyngeal carcinoma, was correlated with the presence of metastases [10]. As VEGFs were found to be overexpressed in PTCs [11], NP-1 immunohistochemical staining seems to be a reasonable candidate as a marker for metastases risk stratification.

Other immunohistochemical markers are Minichromosome maintenance proteins (MCMs). Overexpression of MCM isoform-2 protein (MCM2) enabled the identification of cycling cells and also non-cycling cells with a proliferative potential [12]. Several studies have confirmed MCM2 overexpression in neoplastic cells from different anatomical sites, such as the kidney, stomach, and colon. Moreover, MCM2 was shown to be correlated with tumor progression and metastatic spreading in breast, adrenal and lung cancer [12,13,14]. Nonetheless, MCM2 has not yet been validated in papillary thyroid tumor aggressiveness.

The purpose of the present study is to examine the correlation between the immunohistochemical staining of several proliferative markers and the risk of ETE and DM in PTC.

## 2. Material and Methods

The study included patients who underwent total or hemi- thyroidectomy for multinodular goiter or PTC at Bnai-Zion Medical Center. Inclusion criteria: age > 18 years, histological diagnosis of classical PTC or benign thyroid nodule, available paraffin-embedded thyroid tumor and available medical data regarding ETE or DM. Exclusion criteria: known familial cancer syndrome, personal history of cancer or a history of radiation exposure. The Local Ethics Committee approved the study protocol.

Patients were classified into four groups: The first group comprised patients with the histological diagnosis of a benign neoplasm (multi-nodular goiter or follicular adenoma). The second group comprised thyroid-confined PTC patients who had neither ETE nor metastases (PTC + 0). The third group of patients was characterized by PTC with ETE without known metastases (PTC + E), and the fourth group comprised of PTC patients who had CLNM or DM (PTC + M).

### 2.1. Tissue Processing

Surgical tissue was fixed with formalin (4% Formaldehyde). Dehydration was performed through a gradual increase in alcohol concentrations and the tissue was embedded in paraffin to obtain a block. Three-micron thick serial sections were cut and mounted on positively coated slides. Hematoxylin and eosin staining was performed to enable histopathological diagnosis.

### 2.2. Immunohistochemistry

Benchmark Ultra (Ventana Systems, Phoenix, AZ, USA) was used for the automatic staining procedure. Antigen retrieval and exposure were achieved by incubating the slides in TRIS solution at 95–100 °C for 36–64 min. Immunohistochemical staining was performed by incubation for 32 min with the following antibodies: NP-1 [Rabbit anti-Human Neuropilin-1 antibody Clone EPR3113 (ab81321, Abcam, Cambridge, UK) (diluted 1:80)]; MT [Mouse anti-Human Metallothionein antibody Clone E9 (M0639, DAKO, Glostrup, Denmark) (diluted 1:200)], E-Cadherin [Mouse anti-Human E-Cadherin antibody Clone 4A2C7 (180223, Invitrogen, Waltham, MA, USA) (diluted 1:80)], Ki-67 [Mouse anti-Human Ki67 antibody Clone 7B11 (180192z, Life Technologies, Carlsbad, CA, USA) (diluted 1:100)], and MCM2 [Rabbit anti-Human MCM2 polyclonal antibody (ab4461, Abcam) (diluted 1:600)]. The secondary reaction analysis was performed according to the manufacturers recommended protocol iVIEW DAB detection kit (760-091 Ventana Systems, Phoenix, AZ, USA) or OptiVIEW DAB detection kit (760-700 Ventana Systems, Phoenix, AZ, USA). Hematoxylin was used as a counterstain.

### 2.3. Staining Scoring

The labeling index of proliferation markers Ki-LI and MCM2 was calculated manually as the mean percentage of immunopositively stained cell nuclei in ten fields of 100 tumor cells. Scoring of NP-1, MT, and E-Cadherin cytoplasmic staining was performed through evaluation of the relative intensity of 10 microscopic fields. Score 0 represents no staining; 1 represents weak staining; 2 represents intermediate staining and 3 represents strong staining.

### 2.4. Statistical Analysis

The correlation between each marker staining score and pathological diagnosis was assessed. The statistical analysis of the data was performed using IBM SPSS version 21. Frequency (*n*), percentage, means, and standard deviations (SD) were calculated for each of the four study groups. A comparison between the four study groups was made by *t*-test. The Mann–Whitney test evaluated the relationship between positive staining and ETE, and DM. As immunohistology scores and pathological stages were measured in an ordinal scale, the relation between them was tested by Somers’ D test. Statistical significance was considered for *p*-value < 0.05.

## 3. Results

The study included 66 patients; PTC patients all had the classic type of PTC. Demographic data and tumor size among the four clinicopathological diagnosed study groups are presented in Table 1. No statistical difference was found between the four study groups regarding mean, age, and primary neoplasm greater diameter. MT, E-Cadherin and MCM2 staining significantly differentiated between non-malignant and all groups of malignant tumors (Table 2).

Table 3 displays the statistical differences of immunohistochemical staining intensity score between the different PTC groups. While NP-1, MT and MCM2 did not differentiate between PTC groups, significant inverse correlations were found for E-Cadherin expression in comparison of PTC + 0 vs. PTC + M and PTC + E vs. PTC + M (Figure 1, Figure 2, Figure 3 and Figure 4). Ki-LI nuclear staining was significantly lower in PTC + E than PTC + M; however, it did not significantly differentiate between thyroid-confined tumors and invasive or metastatic ones.

## 4. Discussion

The current study assessed several immunohistochemical proliferative biomarkers in 66 cases of benign and PTC tumors, and its aim was to explore correlations between immunohistochemical proliferative markers and PTC aggressiveness. In line with previous reports, the current study revealed that MT and E-Cadherin staining is significantly lower in malignant tumors and can obviously differentiate between benign thyroid neoplasm and PTC [15,16].

The study showed that no staining used had a correlation with tumor ETE. The only marker that correlated with tumor aggressiveness was E-Cadherin, and exhibited a statistically significant negative correlation with tumor metastases but not with ETE. E-Cadherin is located on the surface area of the epithelial cells in contact points with adjacent cells called adherence junctions, and it is one of the most important molecules involved in epithelial tissues cell adhesion. A weaker immunohistochemistry intensity of E-Cadherin, indicating lower protein expression, is assumed to be associated with higher metastatic nature [17]. Loss of E-Cadherin expression indicating poor prognosis has been observed in tumors of the esophagus, lungs, pancreas, cervix, head and neck [18]. BRAF mutation was reported to exist in 35–70% of PTCs, and one mutation called BRAF V600E has been shown to correlate with aggressive characteristics of PTC, including ETE, advanced tumor stage at presentation, CLNM and DM [19]. One study demonstrated that BRAF V600E mutation upregulates the transcriptional repressor Snail with a concomitant decrease in its target E-Cadherin [20]. These data also explain the inverse correlation between E-Cadherin staining level and PTC aggressiveness.

Zhou et al. assessed the E-cadherin expression levels in relation to clinicopathological characteristics of thyroid cancer through a meta-analysis of 46 studies [21]. The findings showed that E-cadherin negative expression was significantly correlated with lymph node metastasis (ORs = 3.21, 95% CI = 1.98–5.20), tumor node metastasis stage of thyroid cancer (ORs = 4.85, 95% CI = 2.86–8.25) and negatively correlated with differentiation (ORs = 0.25, 95%CI = 0.07–0.82), thus, supporting the results of the present study. Of note, Zhou et al. reported that E-cadherin expression did not have a significant association with distant metastasis, while our study showed that this association is present.

In the current study, MT overexpression differentiated between a benign tumor and PTC, but in contrast to a previous report by Wang et al., who showed that MT overexpression in PTCs is correlated with high TNM tumor stage (stages III–IV) and the presence of CLNM [7], MT staining in the present study did not differentiate between thyroid confined tumor and aggressive one. One reason may originate from the diverse scoring method of staining intensity, as Wang et al. used a semiquantitative assessment of immunohistochemical scoring, using the definition of a high and low score, while we used 4 points score (0 to 3).

More intense MCM2 staining in PTC than benign tumors as demonstrated in our study is a novel finding, but we found that MCM2 is unable to differentiate between PTC TNM stages. In their study, Cho et al. have shown the ability of MCM2 to distinguish between thyroid lesions of minimally invasive follicular carcinoma and follicular adenoma, but they did not study that on PTC tumors [22].

In our study, higher level of Ki-LI staining intensity was observed in metastatic tumors than in those with ETE, but it was surprisingly lower in tumors with ETE than benign ones. One should be aware that the majority (>95%) of PTCs do not demonstrate increased mitoses, and Ki-LI staining may be negative in specimens with tumor necrosis. Besides, it is well known from other tumors that there is tumor diversity in Ki-LI staining, and relatively low and high-labeled areas may coexist within a single tumor. These pitfalls may explain the inconsistent results of the role of Ki-LI in predicting the prognosis of PTC patients [6]. According to our study, NP-1 staining seems to have no role in differentiating benign from PTC tumors nor in differentiating different PTC TNM stages.

The limitations of the study are its small sample size and its retrospective design, lacking long-term patient surveillance. Moreover, we did not have a double reading of the slides, as intra- and inter-observer variability are frequent in staining evaluation.

Guidelines criteria for radioactive iodine treatment in PTC patients are not straightforward and many patients may receive unnecessary treatment [23]. The presence of immunohistochemical staining that can correlate with cancer aggressiveness, may guide clinicians to make the right decision regarding the most appropriate treatment [19]. Studies on larger series of patients are required to validate the data obtained here and to evaluate E-Cadherin staining on long-term prognosis.

## 5. Conclusions

In conclusion, immunohistochemical staining for Metallothionein, E-Cadherin and MCM2 significantly distinguished between benign thyroid tumors and PTC. No staining correlated with ETE, but E-Cadherin staining significantly and inversely correlated with the presence of metastases, suggesting that E-cadherin expression might be a potential predictive factor for disease progression in thyroid cancer and might have a role in clinical decision-making regarding follow-up and the need for additional treatments for PTC.

## Figures and Tables

**Figure 1 medicina-59-00110-f001:**
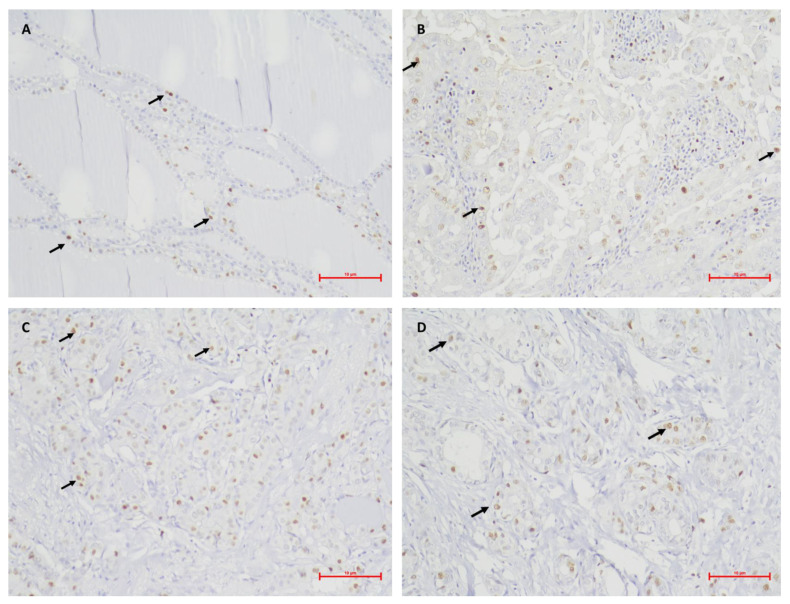
Photomicrographs of the immunohistochemical staining of MCM2 (arrows) in the different studied groups. (**A**), Benign neoplasm, 10% of cells stained (original magnification × 200); (**B**), PTC + 0, 20% of cells stained (original magnification × 200); (**C**), PTC + E, 25% of cells stained (original magnification × 200); (**D**), PTC + M, 25% of cells stained (original magnification × 200).

**Figure 2 medicina-59-00110-f002:**
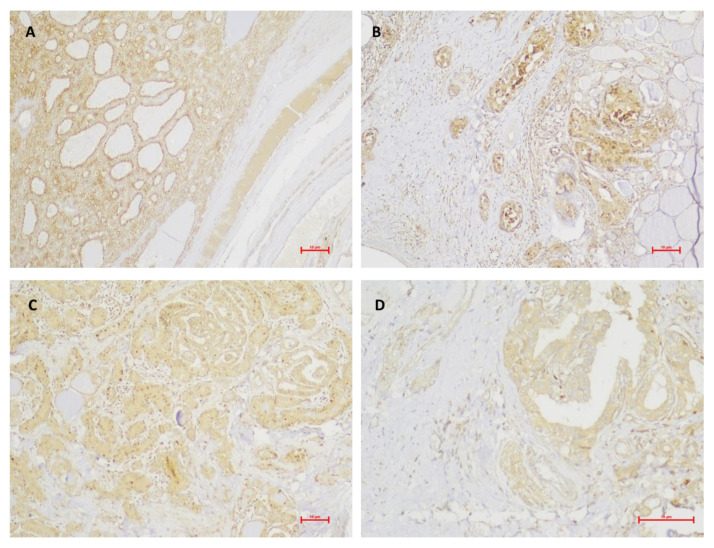
Photomicrographs of the immunohistochemical staining of NP-1 (stained in brown) in the different studied groups. Different groups had the same staining intensity (score 2). (**A**), Benign neoplasm (original magnification × 100); (**B**), PTC + 0 (original magnification × 100); (**C**), PTC + E, (original magnification × 100); (**D**), PTC + M, (original magnification × 200).

**Figure 3 medicina-59-00110-f003:**
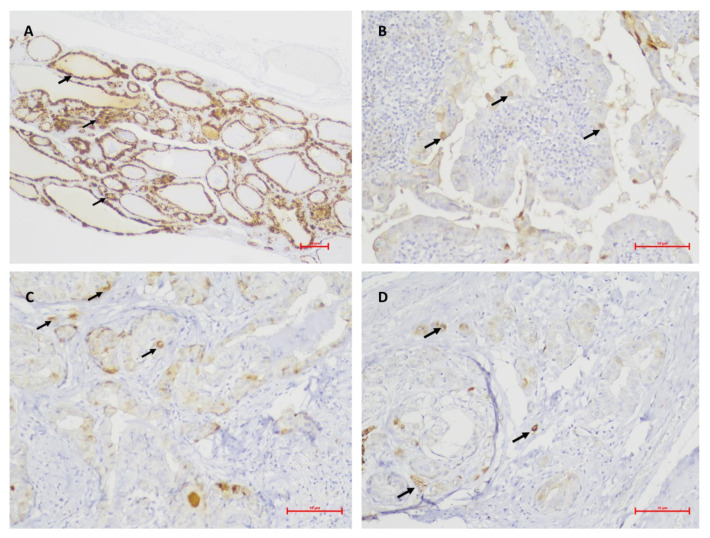
Photomicrographs of the immunohistochemical staining of MT (arrows) in the different studied groups, being able to differentiate benign neoplasm from all PTC groups but not between PTC groups. (**A**), Benign neoplasm, score 3 (original magnification × 100); (**B**), PTC + 0, score 1 (original magnification × 200); (**C**), PTC + E, score 1 (original magnification × 200); (**D**), PTC + M, score 1 (original magnification × 200).

**Figure 4 medicina-59-00110-f004:**
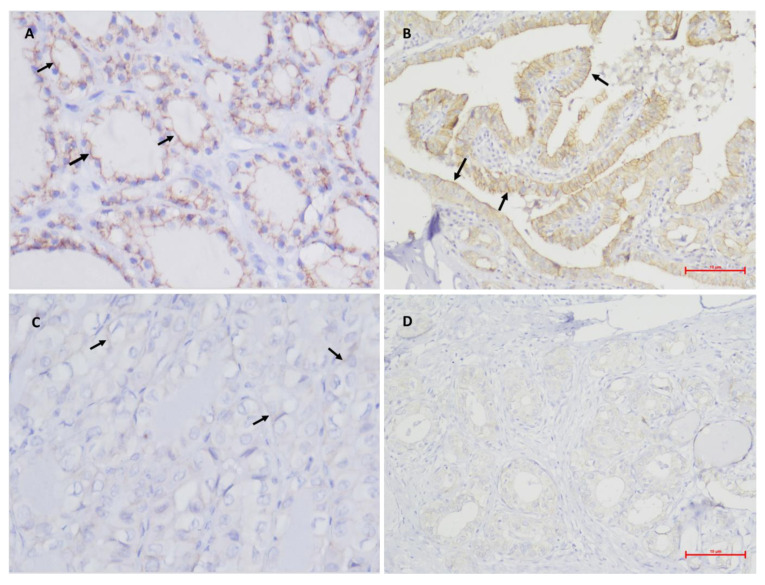
Photomicrographs of the immunohistochemical staining of E-Cadherin (arrows) in the different studied groups, being able to differentiate between all groups. (**A**), benign neoplasm, score 3 (original magnification × 400); (**B**), PTC + 0, score 2 (original magnification × 200); (**C**), PTC + E, score 1 (original magnification × 400); (**D**), PTC + M, score 0 (original magnification × 200).

**Table 1 medicina-59-00110-t001:** Patients’ demographics and neoplasm characteristics.

	Benign Neoplasm	PTC + 0	PTC + E	PTC + M	All Patients
Number of cases	22	21	12	11	66
Age (years)	44.55 ± 14.74	53.14 ± 13.75	59.33 ± 12.13	51 ± 22.33	51.05 ± 16.07
Male/Female	7/15	6/15	3/9	3/8	19/47
Diameter of primary neoplasm (cm)	2.99 ± 1.64	1.33 ± 0.96	1.58 ± 0.57	3.26 ± 1.76	2.25 ± 1.57

Data are presented as mean ± SD.

**Table 2 medicina-59-00110-t002:** Immunohistochemistry staining intensity score of the studied proliferative biomarkers.

	Benign Neoplasm	PTC + 0	PTC + E	PTC + M
NP-1 (score range 0–3)	1.95 ± 1.046	2.1 ± 0.944	2.25 ± 0.45	1.91 ± 0.7
MT (score range 0–3)	2.82 ± 0.66	1.76 ± 1.338 *	1.58 ± 1.24 *	1.27 ± 1.42 *
E-Cadherin (score range 0–3)	1.64 ± 0.727	1 ± 0.837 *	1 ± 0.603 *	0.45 ± 0.82 *
Ki-LI (%)	0.82 ± 1.5	0.62 ± 1.564	0.08 ± 0.29 *	0.86 ± 0.98
MCM2 (%)	11.05 ± 10.67	24.76 ± 22.365 *	24.96 ± 17.84 *	22.55 ± 18.19 *

Data are presented as mean ± SD. * *p*-value < 0.05 in comparison to benign neoplasm.

**Table 3 medicina-59-00110-t003:** Comparisons of immunohistochemical staining intensity score in-between the different studied groups.

	Benign Neoplasm Versus PTC + 0	Benign Neoplasm Versus PTC + E	Benign Neoplasm Versus PTC + M	PTC + 0 Versus PTC + E	PTC + 0 Versus PTC + M	PTC + E Versus PTC + M
NP-1	NS	NS	NS	NS	NS	NS
MT	0.002	0.001	0.01	NS	NS	NS
E-Cadherin	0.004	0.006	<0.001	NS	0.028	0.021
Ki-LI	NS	0.015	NS	NS	NS	0.017
MCM2	0.005	0.01	0.037	NS	NS	NS

Data are presented as *p*-values. For E-Cadherin, r = −0.431. NS: not significant.

## Data Availability

The data are available on request from corresponding author.

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
