# Peer review of "The Correlation between Proliferative Immunohistochemical Markers and Papillary Thyroid Carcinoma Aggressiveness"

_medicina, 2023, doi:10.3390/medicina59010110_

Round 1

Reviewer 1 Report (Previous Reviewer 1)

I am satisfied with the changes the authors have made.

Reviewer 2 Report (Previous Reviewer 4)

I have no further revisions to suggest and the paper may be accepted for publication. 

This manuscript is a resubmission of an earlier submission. The following is a list of the peer review reports and author responses from that submission.

Round 1

Reviewer 1 Report

In this manuscript Sheikh-Ahmad and colleagues investigate the prognostic utility of MCM2, Ki-67, E-Cadherin, Neuropilin-1 and Metallothionein immunohistochemical staining for papillary thyroid carcinoma.

Clinical Significance: The main clinical challenge of thyroid cancer is to avoidance overdiagnosis and overtreatment of patients with low-grade/benign disease, whilst rapidly identifying aggressive cases where the prognosis is usually dismal. Clinical studies and basic research show BRAF and TERTp mutations etc. act to promote more aggressive thyroid cancer. However, there is still debate around their exact clinical utility as prognostic markers, and there is a definite need in our field for the identification of new biomarkers.

Overall, this is a well-designed retrospective study, which has employed an acceptable sample-size and appropriate statistical analyses. I do have some concerns over the minimal IHC data shown, but if the authors can address my concerns, then I believe the study may be suitable for publication in this journal.

Comments:

Numerous studies have investigated the clinicopathological significance of E-cadherin expression in thyroid cancer. See the meta-analysis by Zhou et al 2019. The authors should acknowledge and discuss their data in the context of this large body of published work.

Please provide representative images for all stains tested.

Images should include scale-bars.

Author Response

We added to the revised manuscript a discussion regarding the results of the meta-analysis by Zhou et al 2019. "Zhou et al. assessed the E-cadherin expression levels in relation with clinicopathological characteristics of thyroid cancer through meta-analysis of 46 studies (21). The findings showed that E-cadherin negative expression was significantly correlated with lymph node metastasis (ORs=3.21, 95% CI=1.98-5.20), tumor node metastasis stage of thyroid cancer (ORs=4.85, 95% CI=2.86-8.25) and negatively correlated with differentiation (ORs=0.25, 95%CI=0.07-0.82), thus, supporting the results of the present study. Of note, Zhou et al. reported that E-cadherin expression did not have significant association with distant metastasis, while our study showed that this association is present."

We also added to "results" detailed photomicrographs with scale-bars of the stains.

Reviewer 2 Report

Dear Authors, the manuscript is both well writtwn and organized.Thyroid carcinoma is the most common cancer of the endocrine system, and its prev-33 alence has increased in the recent years for these reasons to study the expression of novel biomarkers are crucial  for making a more accurate diagnosis.

The manuscript is elligible for pubblication in medine after minor revision

Author Response

We made the changes according to the comments and the following was added to conclusions: "suggesting that E-cadherin expression might be a potential predictive factor for disease progression in thyroid cancer and might have a role in clinical decision-making regarding follow up and the need for additional treatments for PTC”. 

Reviewer 3 Report

Comments for the authors:

Ahmad et al. measured the expression levels of tumor markers in patients with papillary thyroid carcinoma. However, there is lack of clinical significance or mechanism research in the current study.

1.     Results of Table 2 and Table 3 showed similar conclusion, there is no additional information.

2.     The authors claimed that significant inverse correlations were found for E-Cadherin expression in comparison of PTC+0 vs PTC+M and PTC+E vs PTC+M (figure 1). The IHC staining of E-cadherin in PTC+M is missing. The demonstration of inverse correlations needs further statistical analysis. The values of R and P should be shown.

3.     Moreover, the prognostic role of E-cadherin in PTC has been reported in previous studies. Why the authors do not focus on the role of NP-1 or MCM2 in PTC?

4.     The representative graphs of tumor markers scored 0-3 should be shown.

5.     The author should perform that univariate and multivariate logistic regression analyses were used to assess risk factors for metastasis or extrathyroidal extension.

Author Response

Reply to comment 1: The aim of table 3 is to clarify and refine in a more detailed way the differences between groups.

Reply to comment 2: We added the photomicrographs of E-cadherin in PTC+M to manuscript. As the E-Cadherin variable was presented at ordinal scale (0-3) and the pathology was also at ordinal scale we used the Somers' D test to test this relation. We added to results the r and made changes on “Statistical analysis” section.

Reply to comment 3: As we found no significant relation between the two stains (NP-1 and MCM2) and the different groups of PTC (as shown in table 2), we do not elaborate our discussion on the role of these two stains in PTC.

Reply to comment 4:  We added photomicrographs of the stains to the results section.

Reply to comment 5: A univariate and multivariate logistic regression analyses were used to assess risk factors for metastasis or ETE. A diameter of 2.05 cm or more of the neoplasm predicted DM. However, as the primary aim of the study was not to identify risk factors for DM, but to study the ability of different stains to predict aggressiveness of the tumor, and as the number of the studied cases was too small to do regression analysis, we chose not to mention or discuss that.

Reviewer 4 Report

This is a retrospective cohort study aimed at assessing whether immunohistochemical markers correlate with papillary thyroid carcinoma aggressiveness.

The abstract is adequate, and has listed rationale and setting details, alongside the most important findings. The introduction should outline why the study encompassed these exact IHC markers. Partial thyroidectomy needs to be elaborated or replaced with hemi- or total thyroidectomy

The objectives of the study are presented clearly and the introduction section communicates the need for investigating the impact of IHC staining on predicting tumor aggressiveness. However, most of the described IHC targets have already been examined on larger cohorts. Several sentences describing previous studies should be transferred to the discussion section.

However, the methodoloy included a small number of patients, evaluated through tests assesing correlation, but not causality. I would suggest using a binary logistic regression model if a causal connection with ENE and DM is to be evaluated. I would also recommend using a STROBE flowchart and checklist. Inclusion and exclusion criteria need to be clearly stated. A power analysis is mandatory, since the number of patients is low. 

The paper is well written, but suffers from a low sample size, and the conclusions do not flow from the results, sinco no correlation was found, and the study did not contribute much to understanding why some PTCs are more aggressive than others.

Author Response

We replaced partial thyroidectomy by hemi-thyroidectomy.

We added to "material and methods" section:

"Inclusion criteria: age > 18 years, histological diagnosis of classical PTC, available paraffin embedded thyroid tumor and available medical data regarding ETE or DM.

Exclusion criteria: known familial cancer syndrome, personal history of cancer or a history of radiation exposure. "

A univariate and multivariate logistic regression analyses were used to assess risk factors for metastasis or ETE. A diameter of 2.05 cm or more of the neoplasm predicted DM. However, as the primary aim of the study was not to identify risk factors for DM, but to study the ability of different stains to predict aggressiveness of the tumor, and as the number of the studied cases was too small to do regression analysis, we chose not to mention or discuss that.

Round 2

Reviewer 3 Report

There is lack of clinical significance or mechanism research in the current study. 

Reviewer 4 Report

I thank the authors on making the suggested revisions to the paper. I believe the analysis of the results was improved, and argument flow was improved.